# Talking Different Languages: The Role of Plant–Plant Communication When an Invader Beats up a Strange Neighborhood

**DOI:** 10.3390/plants12183298

**Published:** 2023-09-18

**Authors:** Rea Maria Hall, Dimitrije Markovic, Hans-Peter Kaul, Helmut Wagentristl, Bernhard Urban, Nora Durec, Katharina Renner-Martin, Velemir Ninkovic

**Affiliations:** 1Institute of Agronomy, University of Natural Resources and Life Science, 3430 Tulln an der Donau, Austria; hans-peter.kaul@boku.ac.at (H.-P.K.); bernhard.urban@boku.ac.at (B.U.); noradurec@gmx.at (N.D.); katharina.renner-martin@boku.ac.at (K.R.-M.); 2Institute of Botany, University of Natural Resources and Life Science, 1180 Vienna, Austria; 3Department of Ecology, Swedish University of Agricultural Sciences, SE-75007 Uppsala, Sweden; dimitrije.markovic@agro.unibl.org; 4Faculty of Agriculture, University of Banja Luka, 78000 Banja Luka, Bosnia and Herzegovina; 5Experimental Farm Groß-Enzerdorf, University of Natural Resources and Life Sciences, 2301 Groß-Enzersdorf, Austria; helmut.wagentristl@boku.ac.at; 6Institute of Mathematics, University of Natural Resources and Life Science, 1180 Vienna, Austria

**Keywords:** kin recognition, invasive species, crops, adaptation, volatile organic compounds, root exudates, interspecies interaction, intraspecies interaction

## Abstract

Communication through airborne volatile organic compounds (VOCs) and root exudates plays a vital role in the multifarious interactions of plants. Common ragweed (*Ambrosia artemesiifolia* L.) is one of the most troublesome invasive alien species in agriculture. Below- and aboveground chemical interactions of ragweed with crops might be an important factor in the invasive species’ success in agriculture. In laboratory experiments, we investigated the contribution of intra- and interspecific airborne VOCs and root exudates of ragweed to its competitiveness. Wheat, soybean, and maize were exposed to VOCs emitted from ragweed and vice versa, and the adaptation response was measured through plant morphological and physiological traits. We observed significant changes in plant traits of crops in response to ragweed VOCs, characterized by lower biomass production, lower specific leaf area, or higher chlorophyll contents. After exposure to ragweed VOCs, soybean and wheat produced significantly less aboveground dry mass, whereas maize did not. Ragweed remained unaffected when exposed to VOCs from the crops or a conspecific. All crops and ragweed significantly avoided root growth toward the root exudates of ragweed. The study shows that the plant response to either above- or belowground chemical cues is highly dependent on the identity of the neighbor, pointing out the complexity of plant–plant communication in plant communities.

## 1. Introduction

Biological invasions by non-native species are a worldwide phenomenon. They are not only one of the major drivers of the restructuring and malfunctioning of ecosystems [1,2], but they can also have a severe economic impact by negatively affecting agricultural production, animal health, and/or human health [3,4]. Most of the studies that have addressed plant invasions have focused on the characteristics of invasive alien species (IAS), the process of invasion [5], the influence of disturbance on invasion success [6], and the attributes of invaded systems [7,8]. In nature, plants live together in communities composed of numerous species that communicate through a variety of complex mechanisms. To compensate for their sessile life form, plants have evolved various mechanisms to perceive and respond to their neighbors [9,10]. One of these processes is chemical messaging, nowadays called plant–plant communication, which plays an important role in plants’ coexistence. Plants release volatile organic compounds (VOCs) into the surrounding environment, many of which play an important role in chemical interaction, not only between plants and insects, but also among plant individuals [11,12,13]. Plants constantly monitor chemical cues in order to distinguish between essential ones predicting competitive neighbors and those cues that are not important for their own fitness [13,14]. In response to essential chemical cues, a single plant can exhibit a multitude of adaptation responses, including physiological and morphological changes, to optimize its performance [14,15,16].

According to “Kin selection theory”, kin recognition and kin selection can serve as explanations for altruistic behavior toward relatives [17]. Further studies on plant–plant interaction integrated the concept of kin selection theory with competition ecology and showed that kin discrimination from strangers occurs in a wide variety of wild as well as domesticated species [18,19,20]. For instance, plants reduce root allocation when interacting with siblings but not when interacting with strangers [21]. Subsequently, it was shown that plants exhibit less competitive traits when interacting with kin than when interacting with non-kin [22,23]. Consequently, cooperative behavior (that is, actions that benefit the group rather than the individual) and even altruistic behavior (actions that benefit other individuals at the cost of the actor) are more likely to occur between kin than between strangers, as the expression of competitive traits is decreased among kin [20]. The majority of studies investigated the cues of relatedness on the basis of root exudates [18,24,25], whereas studies on the role of VOCs in kin discrimination are still underrepresented. For example, sagebrush individuals (*Artemisia tridentata*) responded more effectively to volatile signals emitted by experimentally wounded close relatives than from strangers, leading to increased herbivory defense in kins [26].

A couple of studies have already pointed out the “communication channels” between different crops [27,28] and native species in communities [9,29]. However, studies on the response mechanisms of crops to VOCs released by IAS and the response of IAS to VOCs emitted by crops are scarce, particularly for common ragweed, *Ambrosia artemisiifolia* (L.) (henceforth, ragweed). This annual herbaceous species is native to North America and is nowadays one of the most troublesome agronomic weeds in Europe and Asia. The combination of this reproductive power, including the longevity of the seeds and its highly adaptable behavior, makes ragweed one of the most important IAS, causing considerable yield losses in agriculture [30,31].

For example, oil seed pumpkin yield losses of 70% were reported when the field was infested with 10 ragweed plants per m^2^ [32]. A yield reduction of 30% at densities of 5 ragweed plants per m^2^ was demonstrated for sunflower and maize [33]. Furthermore, it has been shown that the presence of ragweed can also have an indirect negative impact on crops by disrupting the mutualistic symbiosis between plant roots and soil microbes like nitrogen-fixing rhizobial bacteria. Hall et al. [34] showed that the number of rhizobial nodules and their weight (as an indicator for their nitrogen fixing performance) decreased sharply when soybeans or runner beans were grown together with ragweed, causing an average yield loss of up to 80%.

Other studies have shown that chemical compounds released by the genus *Ambrosia* have a broad spectrum of biological activities with the potential to inhibit the germination and growth of other plant species [35,36,37]. Most of these studies investigated the effects of plant residues in aqueous, hexane, or methanol extracts [38,39,40]. To the best of our knowledge, the direct effects of VOCs and root exudates released by ragweed on specific or conspecific neighboring plants have not been investigated. In order to obtain a better insight into the chemical interactions between ragweed and crops, the aim of this study was to test (1) crop (wheat, soybean, and maize) response to VOCs emitted by aboveground parts of ragweed; (2) ragweed response to VOCs emitted by crops or ragweed itself; and (3) whether crop seedlings are capable of detecting and responding to root exudates of ragweed.

## 2. Results

### 2.1. Exposure Experiment

#### 2.1.1. Aboveground Dry Mass (AGDM)

Exposure to ragweed VOCs had a significant impact on the crop’s aboveground dry mass (AGDM) production. As shown in Figure 1a, wheat plants exposed to volatiles released by ragweed had 35.6% less biomass than the unexposed plants. Soybean plants exposed to volatiles from ragweed showed 41.1% less biomass than unexposed plants (Figure 1b). Also, maize responded to volatiles from ragweed by increasing AGDM by 16.6%. A significant reduction of 21.4% of biomass was also observed in wheat plants exposed to their conspecifics. The difference between unexposed soybean (control) and soybean exposed to a conspecific was not significant (−11.4%). In contrast, the biomass production of maize was promoted by exposure to VOCs from conspecifics, where an increase of 35.3% in AGDM was observed (Figure 1c). However, the AGDM of ragweed was not affected, regardless of whether the plants were exposed to volatiles from conspecific or crop plants (Figure 1d).

#### 2.1.2. Stem and Leaf Mass Fraction

Although there were no differences in SMF and LMF between exposed and unexposed wheat, irrespective of the emitter (Figure 2a and Figure 3a), the SMF of soybean plants exposed to ragweed was 13.9% higher and the LMF was 11.5% lower than that of the unexposed control plants. This effect was not noticed when soybean was exposed to conspecifics (Figure 2b and Figure 3b). Regardless of whether maize was exposed to maize or ragweed, the LMF significantly decreased by an average of 8.9%, and the stem biomass increased by 14.1% (Figure 2c and Figure 3c). Volatiles released from the crops had no significant effect on the SMF of ragweed. When ragweed was exposed to ragweed itself and wheat, we found an increase in the LMF of 11.9% and 14.9%, respectively (Figure 2d and Figure 3d).

#### 2.1.3. Specific Leaf Area (SLA)

Exposure to volatiles released by ragweed had a significant impact on the SLA of all tested crops (Figure 4a–d). Wheat, soybean, and maize showed the highest SLA when they were growing without any neighbors. The lowest SLA in wheat was observed with plants exposed to wheat (Figure 4a); these plants showed a 16.0% lower SLA than the unexposed plants. The SLA of soybean plants exposed to soybean decreased by 26.8% compared to unexposed conspecifics and by 11.4% when exposed to ragweed (Figure 4b). A more unresponsive neighboring effect was observed with maize, which reduced SLA by 13.4% and 12.6% when exposed to maize and ragweed, respectively (Figure 4c). Ragweed plants exposed to conspecifics or maize showed the same SLA as unexposed plants, whereas exposure to wheat and soybean led to a significant increase in SLA of 12.7% and 12.3%, respectively (Figure 4d).

#### 2.1.4. Chlorophyll Content

The presence of ragweed had a significant negative impact on the chlorophyll content of wheat and soybean (Figure 5a,b). Whereas wheat and soybean plants exposed to conspecifics showed no change in their chlorophyll content, exposure to ragweed caused a decrease in chlorophyll of 10.9% in wheat and 12.7% in soybean. There was no effect on chlorophyll content in maize, either when exposed to a conspecific or when exposed to ragweed (Figure 5c). In contrast, the chlorophyll content of ragweed was significantly reduced when a neighbor was present, irrespective of the species (Figure 5d).

#### 2.1.5. Root Choice Experiment

Results of the Chi^2^ test showed that all crops significantly avoided root exudates from ragweed. Eighty percent of the wheat roots prefer to grow in the tube filled with conspecific root exudates rather than in the one with ragweed. The roots of soybean and maize seedlings significantly preferred to grow toward tubes with root exudates from their conspecifics (75% and 70%, respectively) than toward tubes connected to root exudates from ragweed (Figure 6a). Ragweed also showed an avoidance behavior toward the root exudates of conspecifics. Only 25% of the roots grew into tubes connected to ragweed root exudates. Furthermore, there was a clear preference for 90% of the ragweed roots to grow into tubes connected to soybean root exudates (Figure 6b). The roots of 60% of ragweed seedlings went for tubes filled with exudates from wheat, whereas the other 40% chose to grow in the direction of exudates from conspecifics. We found no preference in the growth direction of the ragweed roots when they had a choice between maize and ragweed exudates.

We also found clear differences in the root length of the various plant species. Wheat roots that grew into the ragweed solution were, on average, 20.5% longer than wheat roots growing in the wheat solution. Soybean roots were 35.6% longer when they were in contact with the root exudates of ragweed compared to roots that grew in the soybean solution (Figure 7a). Even though maize roots grew significantly more toward the environment of maize exudates, the root length of maize was not affected, regardless of whether roots grew in maize solution or ragweed solution. This is in contrast to the root growth of ragweed roots. Even though ragweed roots showed no preference behavior between ragweed and maize exudates, those roots that grew into the maize solution were 29.1% longer than those that grew into the exudates of conspecifics. The greatest effect was observed on the roots of ragweed seedlings that had the choice between tap water and the solution with ragweed exudates; 75% of them chose to grow toward water (Figure 7b), while the roots that grew into the ragweed solution were 35.5% longer than those growing in tap water (Figure 7b).

## 3. Discussion

We found clear differences in the adaptation response of crops (wheat, soybean, and maize) to ragweed and vice versa, which underlines the importance of chemical communication between plants in kin selection. Chemical cues released by ragweed can have a significant effect on plants’ adaptation to strange neighbors, including changes in biomass production and biomass allocation. We also showed that root exudates can play a vital role in the early stages of seedlings’ growth. Root exudates enable the juvenile plant roots not only to distinguish between different environments (conspecific vs. strange neighbor), but can also induce significant changes in root growth. Thus, this chemical interaction among plants is an important functional component, not only in natural plant communities but also in agricultural systems.

### 3.1. Crop Growth Response after Exposure to VOCs from Ambrosia and Their Kin (Self-Exposure)

Plant traits change in response to a variety of internal and external cues. This so-called phenotypic plasticity is used by plants to compete, acquire resources, or cope with stress [41]. The types of cues to which some plants are known to respond also include the detection of neighbors, particularly whether those neighbors are kin or strangers [21,42].

Following the kin selection theory and concepts from competition ecology, plants in communities can act in cooperative or altruistic ways [17,18]. Altruism toward relatives increases the fitness of the recipient at the cost of the altruistic kin [18,21]. In contrast, cooperating plants help each other by not expressing competitive behavior, leading to a benefit for the neighboring plant without any cost for the helper. In the present study, the aboveground biomass production of maize could be particularly related to cooperative behavior, as biomass production of maize increased significantly when maize was exposed to the volatiles of conspecifics.

In competitive plant communities, the self-beneficial trait of one plant has negative effects on its neighbor since the “selfish” trait increases the focal plant’s performance and decreases neighbor performance [21,43,44]. The volatiles emitted by a conspecific had a significant negative effect on the biomass production of the receiving wheat plants. The decrease in biomass was even higher when wheat was exposed to ragweed. A similar result was observed with soybean, whose biomass was significantly reduced when exposed to the volatiles of ragweed but not when exposed to a conspecific. These results on wheat and soybean are in accordance with the findings of numerous studies, indicating that kin selection will increase the competitive effect between unrelated neighbors and decrease the competitive effect with related neighbors [18,21,45].

It has been shown that traits related to plant growth performance, like SLA, can be linked to their subjective “stress situation” [46]. For example, under low levels of stress (moderate or low competition = lower altruism), plants tend to achieve longer average leaf life spans, which requires extra structural strength, causing lower leaf growth rates and therefore lower SLA [47,48]. A significantly lower SLA in wheat after exposure to wheat and no effect after exposure to ragweed indicate a lower stress in wheat that was exposed to conspecifics. Both SLA and LMF significantly decreased in maize, regardless of whether the emitter was ragweed or a conspecific; this shows that the presence of a neighbor is enough to adjust its growth pattern.

When changing perspective, one might assume that “bad news” is still better than “no news”, as the highest level of stress in this study was experienced by the unexposed plants as they had no information about their neighboring plants, leading to an increase in SLA. To the best of our knowledge, no study has been carried out on this subject. However, this increased SLA is also a major trait of ruderal pioneer plant species, which often grow alone in early successional states [46]. In the present study, this phenomenon was confirmed by all of the crop species that showed the highest SLA when plants were unexposed.

### 3.2. Ragweed Growth Response after Exposure to VOCs from Crops and Its Kin (Self-Exposure)

Exposure of ragweed to VOCs from one of the three crops or to VOCs from conspecifics did not cause changes in biomass production, indicating a broad scale of “antisocial behavior” of invasive plant species. This was particularly confirmed by the SLA, which remained unaffected when ragweed plants were unexposed. This indicates that ruderal plants are adapted to grow alone in newly established habitats [49]. In order to avoid negative effects on their development, pioneer plants like common ragweed can quickly adapt to neighbors (crops and kins) compared to crops [50]. In this context, we found significant adjustments in the LMF of ragweed when exposed to a conspecific, assuming that the VOCs emitted from a kin act as early warning cues pointing to the presence of competition for available resources. Previous studies showed that plants grown with kin neighbors tended to produce more branches and/or leaves as a response to avoid future mutual shading [20,23].

Among crops, exposure to volatiles released by wheat had the highest stress effect on common ragweed, causing a significant increase in LMF and SLA. Among all of the plant species included in the study, the ecological and physiological gap between wheat and common ragweed is the widest; this underlines Hamilton’s rule [17] that predicts that unrelated individuals are less likely to cooperate. Annual ruderal pioneer species like ragweed usually have a comparably low competitive power against vigorous monocotyledonous *Poaceae* species like wheat [32,50]. Furthermore, grass species in particular are more likely to increase their competitive traits in non-kin interactions [51], whereas dicotyledonous species show a much broader spectrum of reactions [20,52,53].

### 3.3. Changes in Crop Chlorophyll Content after Exposure to VOCs from Conspecifics and Ragweed

It is possible that certain allelochemicals (depending on their type and concentration) may enhance or reduce the synthesis of chlorophyll pigments [54,55,56]. Aqueous extracts of *Tithonia diversifolia*, *Malva parviflora*, and *Chenopodium murale* led to a decrease in chlorophyll content in maize [57,58]. Root extracts of sunflower caused a significant increase in chlorophyll in different wheat varieties (Kamal 2011). Han et al. reported a significant decrease in chlorophyll when various weedy species were treated with the essential oil of ragweed [56]. The present study revealed that the presence of ragweed volatiles can have a severe impact on the chlorophyll content of receiver plants. While the chlorophyll content of maize did not change after exposure to volatiles emitted by ragweed, the chlorophyll content of wheat and soybean was significantly reduced, whereas exposure to a conspecific did not have any impact. SLA and chlorophyll content are usually positively related, so an increased SLA is often accompanied by a higher chlorophyll content [59,60]. This is confirmed by the results we observed for wheat (r^2^ = 0.46; *p* = 0.012, results not shown) and ragweed (r^2^ = 0.65; *p* < 0.001, results not shown), which showed the highest chlorophyll content when grown without any neighboring plants. In general, chlorophyll synthesis is dependent on a series of enzymatic reactions that are sensitive to various environmental factors like nutrient, light, and water availability [61,62]. However, it was shown that variations in the chlorophyll content among coexisting species are significantly more influenced by the community composition than by site conditions like soil and weather, indicating a strong influence of plant–plant communication [63].

### 3.4. Root Seedlings Preference/Avoidance Behavior in Response to Ragweed or Conspecifics

The root choice experiment clearly showed that the roots of all crop seedlings had a significant avoidance behavior against root exudates of ragweed. Furthermore, with wheat and soybean, we found a significant impact on root length, as both crop species showed an increase in root length when they were growing in the presence of root exudates from ragweed. The first evidence for the role of exudates in kin recognition was provided by Biedrzycki et al., who found stronger root growth in *A. thaliana* seedlings when exposed to exudates from strangers’ roots than from conspecifics [64]. Other studies on various species confirmed these results, showing that seedlings produced not only more root mass but also higher specific root length when grown in solutions with exudates from non-relatives than in solutions with exudates from kins [22,25].

It can be assumed that, particularly in the early development stages of plants, increased root length suggests that the presence of competitors triggers this root growth to pre-empt resources from neighbors [10,14,65,66]. This highly specific neighbor recognition in plants was also observed with invasive species like *Centaurea maculosa* and *Thymus pulegioides*, which caused a significant increase in root growth in various native herbaceous species [2]. As communication is not a one-way street, ragweed root growth was also enhanced by the root exudates of conspecifics in comparison to roots that grew in tap water. However, we found significant avoidance behavior against root exudates from kins and a clear preference of ragweed seedlings for root exudates from soybean. As soybean is a comparably weak aboveground and belowground competitor to ragweed [34], it is suggested that soybean exudates are preferred by ragweed seedlings when given the choice between root exudates of kins or soybean. In general, ragweed avoids growing in a competitive environment, but when it is confronted with potential competitors, it triggers a response to pre-empt resources before the competitor [65,67], which was confirmed by significantly higher root lengths when grown in the direction of exudates from conspecifics or maize.

## 4. Material and Methods

### 4.1. Plant Material

We used wheat (*Triticum aestivum*, cv. Capo), soybean (*Glycine max*, cv. Angelica), and maize (*Zea mays*, cv. ES Katamaran) as model plants, all of which were obtained from Saatzucht Probstdorf GesmbH, Probstdorf, Austria. Seeds of ragweed were harvested in autumn 2019 on five different sites in Austria and mixed together to avoid possible parental or habitat/site effects. Before sowing, the seeds were germinated in Petri dishes on filter paper. Immediately after germination (radicle visible), the seedlings were transferred into (9 × 9 × 7 cm) cm pots filled with potting soil. Plants were grown with one seed per pot in a growing chamber at 18–22 °C, with a photoperiod of 16:8, a light intensity of 150 μmol m^−2^ s^−1^ (HQIE lamps—Hortilux Schreader, HPS 400 Watt, Dan Haag, The Netherland), with one lamp per square meter), and a relative humidity of 60 ± 10%, and watered via an automated water drip system. No fertilizers were given to plants during the experimental period. The plants were transferred to the exposure system when the first leaf was fully developed.

### 4.2. Exposure System

In the first run, the exposure of the crops to the volatiles of a conspecific or ragweed was performed in a series of “twin-chamber cages” to ensure that the only source of information for the plant kin recognition were the volatile organic compounds emitted by the various neighboring plants [10,11,66,68].

As shown in Figure 8, air entered the front inducing chamber with an emitter plant through an opening (7 cm in diameter) and passed through the hole in the middle wall into the chamber with a receiving plant before being vented outside the room by a vacuum pump. The airflow through the cages was 1.3 l min^−1^. To avoid volatile interaction before starting the exposure experiment, all plant species were grown in separate growing chambers until the first leaves were fully developed at the first node, as described above. Pots with emitting and receiving plants were placed in separate Petri dishes to prevent potential interaction between plants by root exudates. Then, plants were transferred into the cages in 14 replications of each pairing (ragweed exposed to the crop, crop exposed to the crop, and crop exposed to clean air). In the second experimental run, ragweed was exposed to itself and the crops. As a control, 14 seedlings of all plant species were exposed to clean air (the emitter chamber was empty, so they did not have any VOC information). At the beginning of the exposure experiments, all plant species were of the same age (first leaves fully developed). Subsequently, they were left for 21 days in the exposure chambers. Both runs of the exposure experiment were executed in the lab of the Department of Ecology at the Swedish University of Agricultural Sciences.

### 4.3. Measurements

After 21 days of exposure, the plants were harvested. Before harvest, the chlorophyll content of all plants was measured using the non-destructive chlorophyll meter SPAD-502 Plus (Konica Minolta Sensing Europe B.V, Nieuwegein, The Netherlands), which determines the relative quantity of chlorophyll present in leaf tissue by measuring the transmittance of the leaf in the red and infrared spectral regions [69].

Directly after cutting, the leaf surface area of individual plants was measured using WinRHIZO Version 4 image analysis software (Regent Instruments Inc., Quebec, Canada). Leaves and stems from each plant were separately packed into labeled aluminum bags and dried for 48 h at 70 °C to a constant dry mass. All samples spent 24 h at room temperature before they were weighed. These data were used for calculating the aboveground dry matter (AGDM, stem + leaves) and the morphological indices, namely the leaf mass fraction (LFM)—proportion of leaf mass to AGDM—and the stem mass fraction (SMF)—proportion of stem mass to AGDM. Specific leaf area (SLA) is calculated as the proportion of leaf surface to leaf dry weight.

### 4.4. Root Choice Experiment

A root choice experiment was conducted in the laboratory of the University of Natural Resources and Life Science Vienna following the protocol of Elhakeem et al. [10]. In this experiment, we tested the ability of the roots of germinated crop seedlings to choose between two spatial growth niches that contained root exudates of either a conspecific or ragweed. To obtain the root exudates, seeds of all crops and ragweed were germinated in Petri dishes with light cycle intervals of 12 h and a temperature of 20 °C. During the dark phase, the temperature dropped to 15 °C. When the radicle was approx. 2 cm long, seedlings were transferred to glassy laboratory buckets (500 mL) filled with half-strength Hoagland solution (HH2395-10L, Sigma-Aldrich) dissolved in distilled water. We prepared two buckets for each plant species in order to gain 1000 mL of root exudate for each one. To prevent volatile interactions between plants in neighboring buckets, they were separated into different growing chambers.

After 14 days, the plants were removed, and the buckets were sealed off until the start of the main experiment. For that, we again pre-germinated seeds of all crops and ragweed in Petri dishes until the radicle and the cotyledons were visible. Then, each seed was placed inside the upper opening (1 cm in diameter) of an inverted Y-shape tube. All inverted Y-shape tubes were lined with filter paper and fixed into two 15 mL conical centrifuge tubes from the bottom openings (Figure 9).

One tube was filled with the solution of the conspecific crop, and the other was filled with the ragweed solution. All parts were then fixed vertically on the outer wall of Perspex cages, solely for the purpose of support. Cages were covered with black plastic to provide darkness to the roots and placed in separate growing chambers to avoid possible bias through aboveground interaction between plants. The direction (choice) of the main root of each seedling was recorded seven days after seeds had been placed inside the inverted Y-tubes. Forty seedlings from each crop were given the choice between conspecific and ragweed solutions. Forty seedlings of ragweed always had to choose between the various crop solutions: ragweed and tap water. After 10 days, the Y-tubes were opened, the direction in which the roots grew was verified, and the length of the main root axis from seed to tip was recorded.

### 4.5. Statistical Analysis

Data analysis was performed using R software, Version 3.4.4 (R Core Development Team, 2018). For the graphical visualization of the results, we used the R software and SigmaPlot, Version 14.0 (Systat Software, 2018). Analyses of variance with subsequent multiple comparisons of means according to Tukey were performed (significance level α = 0.05). The Shapiro–Wilk test was used to test the normal distribution of data, and Levene’s test was used to check the homogeneity of variances. If a normal distribution was not found, a Kruskal–Wallis ANOVA on ranks was performed. If homogeneity of variances was not found, a statistical analysis was executed using Welsh’s unequal variances *t*-test. In the root choice test, a Chi^2^-test was performed to check if the distribution of roots was random or if a significant pattern was observable.

## Figures and Tables

**Figure 1 plants-12-03298-f001:**
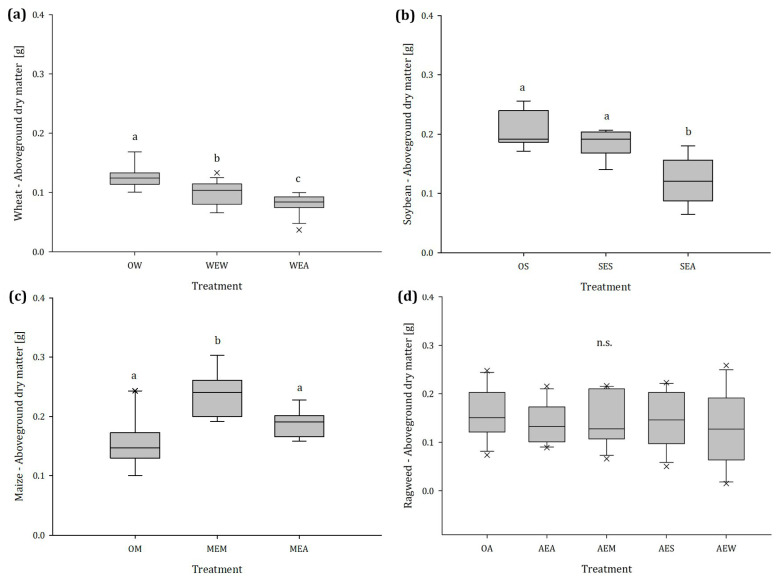
Aboveground dry mass of (**a**) wheat, (**b**) soybean, (**c**) maize, and (**d**) ragweed in dependence of the emitting plant species, n = 14; different letters indicate significant differences, n.s. no statistically significant differences were detected among treatments. Abbreviations: OW = unexposed wheat, WEW = wheat exposed to wheat, WEA = wheat exposed to ragweed, OS = unexposed soybean, SES = soybean exposed to soybean, SEA = soybean exposed to ragweed, OM = unexposed maize, MEM = maize exposed to maize, MEA = maize exposed to ragweed, OA = unexposed ragweed, AEA = ragweed exposed to ragweed, AEM = ragweed exposed to maize, AES = ragweed exposed to soybean, AEW = ragweed exposed to wheat.

**Figure 2 plants-12-03298-f002:**
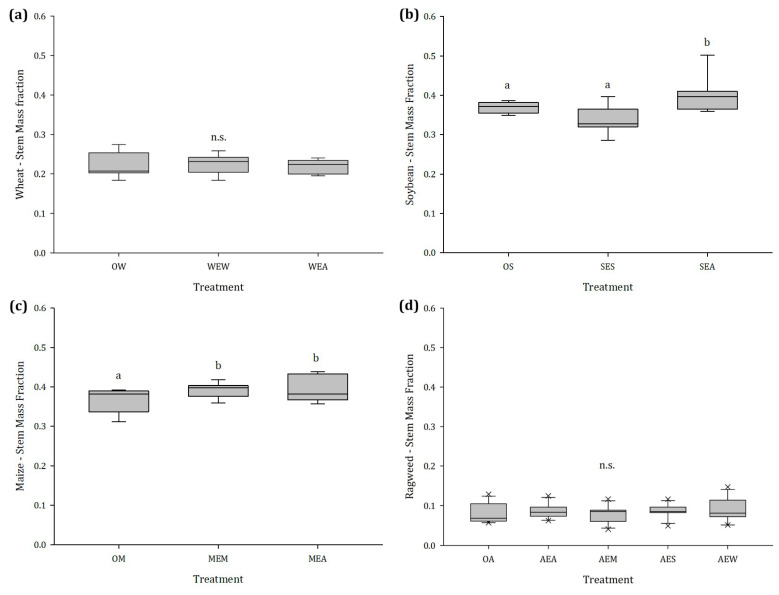
Stem mass fraction of (**a**) wheat, (**b**) soybean, (**c**) maize, and (**d**) ragweed in dependence of the emitting plant species, n = 14; different letters indicate significant differences, n.s. no statistically significant differences were detected among treatments. Abbreviations: OW = unexposed wheat, WEW = wheat exposed to wheat, WEA = wheat exposed to ragweed, OS = unexposed soybean, SES = soybean exposed to soybean, SEA = soybean exposed to ragweed, OM = unexposed maize, MEM = maize exposed to maize, MEA = maize exposed to ragweed, OA = unexposed ragweed, AEA = ragweed exposed to ragweed, AEM = ragweed exposed to maize, AES = ragweed exposed to soybean, AEW = ragweed exposed to wheat.

**Figure 3 plants-12-03298-f003:**
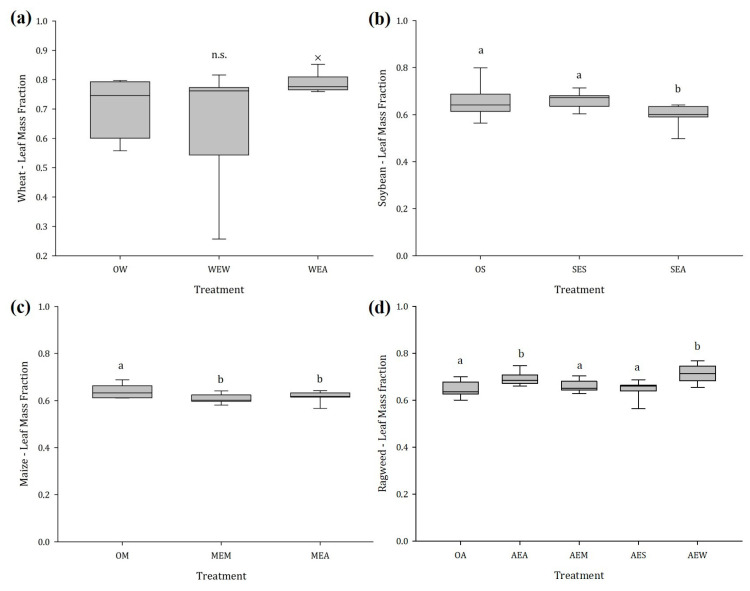
Leaf mass fraction of (**a**) wheat, (**b**) soybean, (**c**) maize, and (**d**) ragweed in dependence of the emitting plant species, n = 14; different letters indicate significant differences, n.s. no statistically significant differences were detected among treatments. Abbreviations: OW = unexposed wheat, WEW = wheat exposed to wheat, WEA = wheat exposed to ragweed, OS = unexposed soybean, SES = soybean exposed to soybean, SEA = soybean exposed to ragweed, OM = unexposed maize, MEM = maize exposed to maize, MEA = maize exposed to ragweed, OA = unexposed ragweed, AEA = ragweed exposed to ragweed, AEM = ragweed exposed to maize, AES = ragweed exposed to soybean, AEW = ragweed exposed to wheat.

**Figure 4 plants-12-03298-f004:**
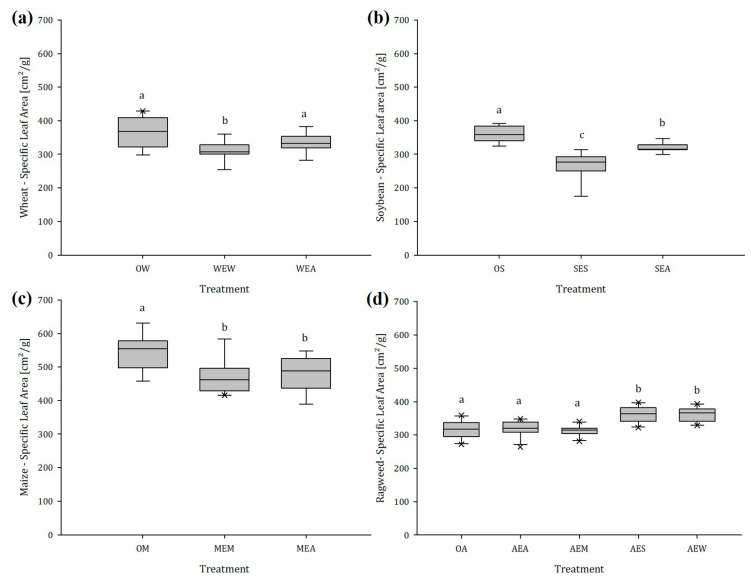
Specific leaf area of (**a**) wheat, (**b**) soybean, (**c**) maize, and (**d**) ragweed in dependence of the emitting plant species, n = 14; different letters indicate significant differences. Abbreviations: OW = unexposed wheat, WEW = wheat exposed to wheat, WEA = wheat exposed to ragweed, OS = unexposed soybean, SES = soybean exposed to soybean, SEA = soybean exposed to ragweed, OM = unexposed maize, MEM = maize exposed to maize, MEA = maize exposed to ragweed, OA = unexposed ragweed, AEA = ragweed exposed to ragweed, AEM = ragweed exposed to maize, AES = ragweed exposed to soybean, AEW = ragweed exposed to wheat.

**Figure 5 plants-12-03298-f005:**
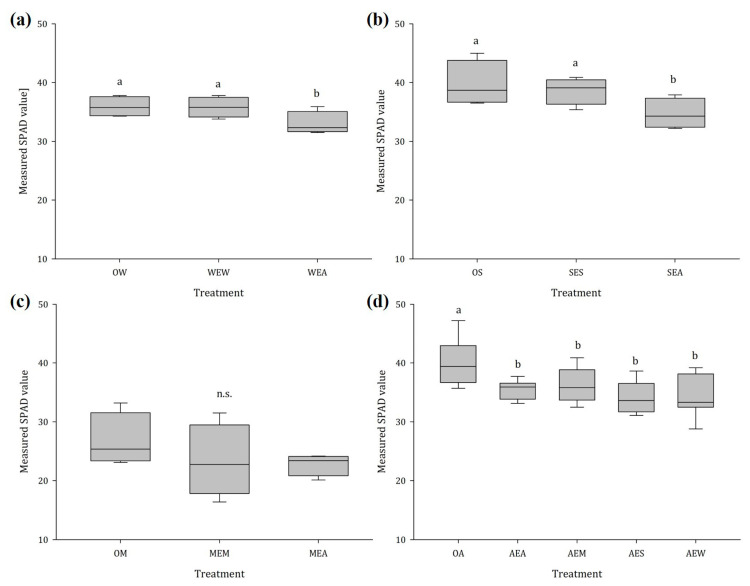
Chlorophyll content of (**a**) wheat, (**b**) soybean, (**c**) maize, and (**d**) ragweed in dependence of the emitting plant species, n = 14; different letters indicate significant differences, n.s. no statistically significant differences were detected among treatments. Abbreviations: OW = unexposed wheat, WEW = wheat exposed to wheat, WEA = wheat exposed to ragweed, OS = unexposed soybean, SES = soybean exposed to soybean, SEA = soybean exposed to ragweed, OM = unexposed maize, MEM = maize exposed to maize, MEA = maize exposed to ragweed, OA = unexposed ragweed, AEA = ragweed exposed to ragweed, AEM = ragweed exposed to maize, AES = ragweed exposed to soybean, AEW = ragweed exposed to wheat.

**Figure 6 plants-12-03298-f006:**
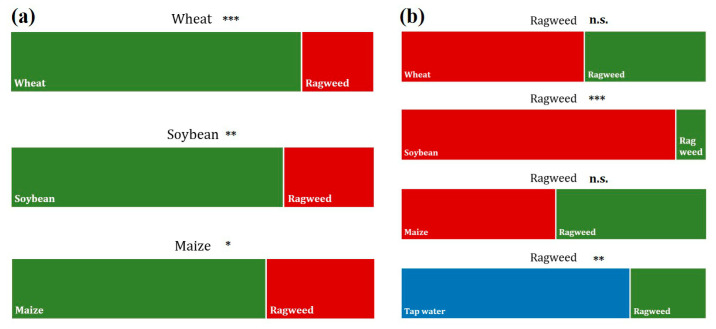
Root growth behavior of (**a**) crops (wheat, soybean, and maize) when exposed to root exudates of conspecifics or ragweed and (**b**) ragweed when exposed to root exudates of conspecifics, crops (wheat, soybean, and maize), or tap water; n = 40; significance levels: * *p* < 0.05, ** *p* < 0.01, *** *p* < 0.001, n.s. no statistically significant differences among treatments.

**Figure 7 plants-12-03298-f007:**
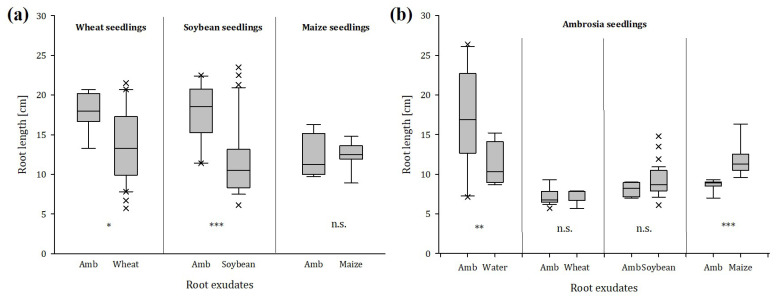
Root length of (**a**) crops (wheat, soybean, and maize) when exposed to root exudates of conspecifics and ragweed (Amb) and (**b**) ragweed when exposed to root exudates of conspecifics, crops (wheat, soybean, and maize), or tap water; n = 40; significance levels: * *p* < 0.05, ** *p* < 0.01, *** *p* < 0.001, n.s. no statistically significant differences among treatments.

**Figure 8 plants-12-03298-f008:**
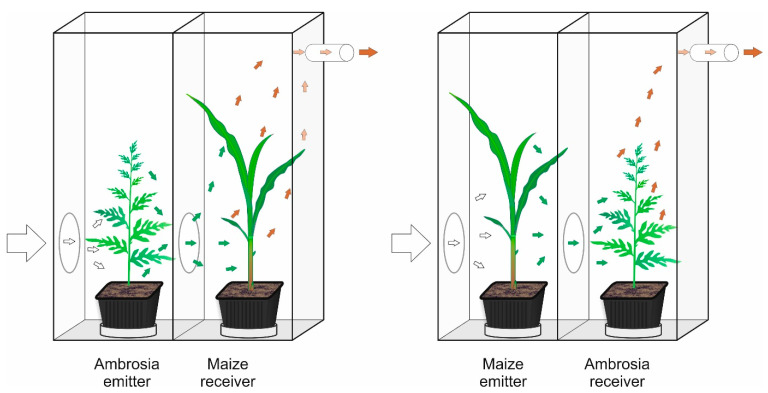
System of twin-chamber cages for exposure of one plant to volatiles from another. The responding plant (receiver) was exposed to volatiles emitted from an inducing plant (emitter).

**Figure 9 plants-12-03298-f009:**
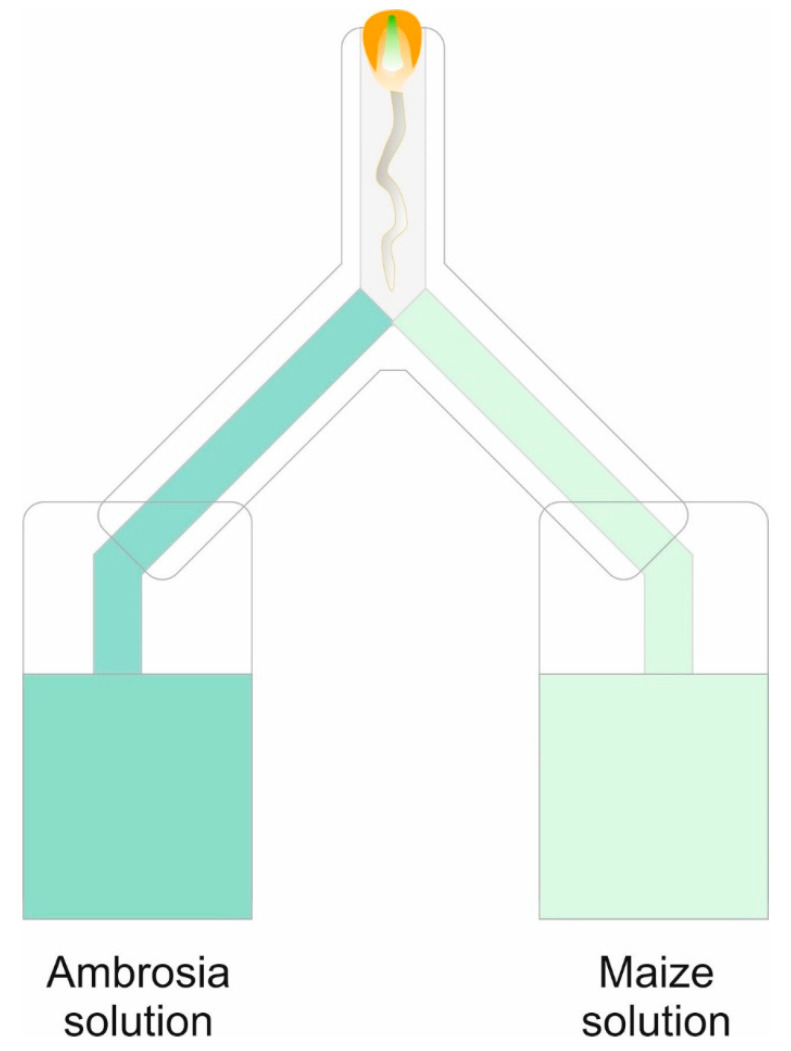
Root choice test in inverted Y-tubes where the roots of crop seedlings had the choice between growth solution with root exudates of a conspecific or ragweed; ragweed was given the choice between root exudates of conspecifics or the three different crops (wheat, soybean, and maize).

## Data Availability

Due to privacy restrictions, we cannot publish the raw data. If the data are needed for further research, we are willing to send them on request. Therefore, please send a short statement to the corresponding author explaining why and for what purpose the data are needed.

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
