# Peer review of "Talking Different Languages: The Role of Plant–Plant Communication When an Invader Beats up a Strange Neighborhood"

_plants, 2023, doi:10.3390/plants12183298_

Round 1
Reviewer 1 Report
The paper "Talking different language: the role of plant-plant communication when an invader beats up a strange neighborhoog" aims to define the impact of VOC and roots exudates from common rageweed on three crop species (maize, soybean and wheat). This impact was evaluated through biomass production, chlorophyll content and morphological traits. This paper is quite novel and very interesting. It is also well written.
My major concern is about SLA. I think the authors made a confusion between specific leaf area (SLA, cm².g-1) and the leaf mass per area (LMA, g.cm-2). According to the figure 4, it seems that they calculated LMA (g.cm-2). However, in the results and discussion, they always talked about SLA. Even though, those two parameters are quite close, they indicate opposite results. Indeed, a low SLA indicates a high stress situation with thick leaves (meaning a high production of specialized metabolites usually) whereas a low LMA indicates a low stress situation with thin leaves. So confusing both parameters changes totally the discussion. It needs to be clarify throuhouot the manuscript. For more informations about LMA, please check Pooter et al. 2009 Causes and consequences of variation in LMA, a meta-analysis.
Then, have you done any check to be sure that the receiver really got the BVOC of the emitter? Was this system equipped with a pump?
Finally, have you done any characterization of BVOC and root exudates. If yes, would be worth to add it in supplementary materials.
Author Response
The paper "Talking different language: the role of plant-plant communication when an invader beats up a strange neighborhoog" aims to define the impact of VOC and roots exudates from common rageweed on three crop species (maize, soybean and wheat). This impact was evaluated through biomass production, chlorophyll content and morphological traits. This paper is quite novel and very interesting. It is also well written.
Thank you very much for this positive feedback and your very valuable comments.
My major concern is about SLA. I think the authors made a confusion between specific leaf area (SLA, cm².g-1) and the leaf mass per area (LMA, g.cm-2). According to the figure 4, it seems that they calculated LMA (g.cm-2). However, in the results and discussion, they always talked about SLA. Even though, those two parameters are quite close, they indicate opposite results. Indeed, a low SLA indicates a high stress situation with thick leaves (meaning a high production of specialized metabolites usually) whereas a low LMA indicates a low stress situation with thin leaves. So confusing both parameters changes totally the discussion. It needs to be clarify throuhouot the manuscript. For more informations about LMA, please check Pooter et al. 2009 Causes and consequences of variation in LMA, a meta-analysis.
We apologize so much for this confusion. We initially calculated SLA (cm²/g) but when drawing the graphs we thought of transforming the results to cm² per milligram (mg) but then we chose the first option again. However, during the conversion from gram to milligram and backward an error with the number of “0” occurred. Consequently, the graphs were not proper scaled. In addition, the labelling was wrong... Sorry, for these two faults in the same graph. We mended it all and exchanged the graph. Page 6.
Then, have you done any check to be sure that the receiver really got the BVOC of the emitter? Was this system equipped with a pump?
The airflow through the system was tested with smoke and the air in the first chamber was completely changed passing from emitting to receiving chamber within 2 min and then vented outside of growing chamber.
Yes, the system was equipped with vacuum pump as described in the lines 362-363.
Finally, have you done any characterization of BVOC and root exudates. If yes, would be worth to add it in supplementary materials.
Yes, we did an analyzation of ragweed extracts in another study. Results are already published in:
Hall RM, Wagentristl H, Renner-Martin K, Urban B, Durec N, Kaul, H-P (2023). Extracts and Residues of Common Ragweed (Ambrosia artemisiifolia L.) Cause Alterations in Root and Shoot Growth of Crops. Plants 12, 1768. https://doi.org/10.3390/ plants12091768
https://www.mdpi.com/2223-7747/12/9/1768

Reviewer 2 Report
It is topic which opens wide perspectives in both substantive and highly speculative research of essential oils. Title The title is very mannered, too general, providing no knowledge of the study, more suitable for a review article, where results from the numerous studies have to be analyzed Abstract The summary begins and ends with a vague statements: “...vital role in the multifarious interactions of plants...” and “...pointing out the complexity of plant–plant communication in plant communities...” We know that the interaction is complex from previous abundant studies. It is research article, based on a single study only, that is why among multifarious interactions some exact ways of communications might be disclosed. Information is missing whether present study is field experiment or laboratory assessment. Briefly (in several words) country of investigations (climate, temperature as the most important factors) should be mentioned, also stage of soybean, maize or wheat growth and development when crops, Ambrosia have been treated and/or have been used for extraction of essential oils should be described. Main result quantification is missing: “we observed significant changes in plant traits” We observed significant changes in plant traits – when it has been observed after treatment? Keywords Names of the main investigated species are missing, it decreases probability of the article detection and citing Introduction References 1-4 and 5-7 should be split by processes describedCommunities belong to the attributes of invaded systems Row 39 the use of „However“ logistically is not correct.
It is hard to say which expression is older, duration of the use of both are nearly equal: Rows 42-43
„...force plants to monitor, detect“... – the meaning of the phrase is under question
„Plants release more than 1,700 volatile organic compounds (VOCs)“, taking into account only 4 plant species investigated such sample is not relevant. Data about VOCs of the plants under investigation might be of bigger value.
Kin relation interpretations might be totally different.No Ambrosia artemesiifolia actuality for regions/countries in Europe is provided. No information about Ambrosia seed density in the agrofields and semi-natural ecosystems of the countries. Species specific actuality for the wheat, soybean and maize fields also is not disclosed. All effects depend on numbers of the plants, on mass of neighboring plants, their height and duration of stages of comparable species.
Methods
Appearance of the seedlings is not described (at least height and the number of the leaves). Similarly appearance of the seedlings after exposure is not registered. The best would be to use BBCH scale for plant description. And information is missing why such stages were selected. The same information is missing for the root experiment.
Description of the age of the experimental seedlings is very scattered and confusing, it is too long description. The number of seedlings to which root extract was exposed and how many seedlings were exposed are also unclear.
The shortest and clearest way would be to make a table indicating the age of seedlings and the number of individuals selected for both the smallest experimental unit and the number of seedlings used in general. It is not clear where plants growing in one pot/tray/dish were used and where batches of plants were used.
No information is provided about microbial contamination possible effects in root exudation experiment.
Methodical references are only for barley (11,68).
Eight out of 68 references belong to the authors of the article
Results
Age of the seedlings should be provided in the titles of the tables and the numbers of the plants used in separate replicates and the total number of plants should be provided
Discussions
Present study data are not compared with existing VOC data concerning selected species.
„In general, chlorophyll synthesis is not only dependent on numerous environmental factors like nu- trient, light, and water availability, but also consists of a series of enzymatic reactions that are sensitive to various biotic and abiotic factors“ two parts of the sentence are talking about the same – I would reject such a phrase.
„showed that variations in the chlorophyll content among coexisting species are significantly more influenced by the community composition than by site conditions like soil and weather, indicating a strong influence of plant–plant communication“ might be very speculative interpretation due to limited experimental cases.
References
Should be more related to the species under investigation
In conclusion, present research is pure sterile laboratory experiment, which does not have relation to the reality either in agroecosystems or seminatural habitats. It may serve for governoment speakers to talk about Ambrosia threats.
Some phrases are not clear
Author Response
It is topic which opens wide perspectives in both substantive and highly speculative research of essential oils.
Title The title is very mannered, too general, providing no knowledge of the study, more suitable for a review article, where results from the numerous studies have to be analyzed
Thank you for your suggestion, but we would prefer to keep the title as it is. We are convinced that the title points the main message of our studies.
Abstract The summary begins and ends with a vague statements: “...vital role in the multifarious interactions of plants...” and “...pointing out the complexity of plant–plant communication in plant communities...” We know that the interaction is complex from previous abundant studies. It is research article, based on a single study only, that is why among multifarious interactions some exact ways of communications might be disclosed. Information is missing whether present study is field experiment or laboratory assessment.
Unfortunately, we don’t understand this comment and we are ready to answer on this question after clarification. We included in the abstract that this was a laboratory experiment.
Briefly (in several words) country of investigations (climate, temperature as the most important factors) should be mentioned, also stage of soybean, maize or wheat growth and development when crops, Ambrosia have been treated and/or have been used for extraction of essential oils should be described.
The laboratory experiment was executed at the Swedish University of Agricultural Sciences in Uppsala and University of Natural Resources and Life Science Vienna (Austria) which is irrelevant as both studies were performed in well-equipped university laboratories. The seeds of the crops were obtained from a breeding company. Ambrosia seeds were harvested on infested fields in Austria. All experiments were conducted under optimum laboratory conditions.
Furthermore, there was no treatment in the narrow sense of the word as crops and Ambrosia were only exposed to airborne VOCs of a neighbor.
Essential oil was never mentioned and has not been used in the experiment.
Main result quantification is missing: “we observed significant changes in plant traits” We observed significant changes in plant traits – when it has been observed after treatment?
As described the above, the “treatment” was, that crops were exposed to airborne volatiles of a kin neighbor or Ambrosia and the exposure of Ambrosia to airborne VOCs from a kin or crops, respectively. As control, 14 seedling of all plant species were grown without any neighbor (clean air - no airborne VOC information).
Keywords Names of the main investigated species are missing, it decreases probability of the article detection and citing
The names of investigated species and cultivar names are mentioned in the abstract and also with Latin names of the crops lines 343-344.
Introduction References 1-4 and 5-7 should be split by processes described
Thank you for the suggestion and the changes was done accordingly.
Communities belong to the attributes of invaded systems Row 39 the use of „However“ logistically is not correct.
Thank you for this indication.” However” is deleted Line 41.
It is hard to say which expression is older, duration of the use of both are nearly equal: Rows 42-43
Sorry, but we don’t understand this comment.
„...force plants to monitor, detect“... – the meaning of the phrase is under question
Thank you for this indication. That sentence is rewritten. Line 48-49
„Plants release more than 1,700 volatile organic compounds (VOCs)“, taking into account only 4 plant species investigated such sample is not relevant. Data about VOCs of the plants under investigation might be of bigger value.
Thank you for this indication. That sentence is rewritten. Line 46.
Kin relation interpretations might be totally different.
We agree, but we used the most common interpretation of kin relation in the literature.
No Ambrosia artemesiifolia actuality for regions/countries in Europe is provided. No information about Ambrosia seed density in the agrofields and semi-natural ecosystems of the countries. Species specific actuality for the wheat, soybean and maize fields also is not disclosed. All effects depend on numbers of the plants, on mass of neighboring plants, their height and duration of stages of comparable species.
This information is not relevant for the present study as we investigated only how volatiles of Ambrosia artemisiifolia affect different traits of crops and vice versa in a laboratory experiment.
Methods
Appearance of the seedlings is not described (at least height and the number of the leaves). Similarly appearance of the seedlings after exposure is not registered. The best would be to use BBCH scale for plant description. And information is missing why such stages were selected. The same information is missing for the root experiment.
As described in the Methods section we pre-germinated the seedling of all crops and Ambrosia in petri dishes. Immediately after germination (radicle visible) the seedlings were grown separately (one per pot) until the first leaves were fully developed at the first node (line 347-349 and 405-408). When plants achieved this development the exposure experiment started.
Summarizing: at the beginning of the exposure experiment all plant species were of the same age (first leaves fully developed). Subsequently, they were left for 21 days in the exposure chambers.
The root experiment was similar. Seeds of all species were pre-germinated in petri dishes until the radicle and the cotyledones were visible. Then they were transferred to the tubes.
Description of the age of the experimental seedlings is very scattered and confusing, it is too long description. The number of seedlings to which root extract was exposed and how many seedlings were exposed are also unclear.
In the exposure experiment each pairing was replicated 14 times. Line 359-362
In the root choice test 40 seedlings of each species were tested. Line 415-416
The shortest and clearest way would be to make a table indicating the age of seedlings and the number of individuals selected for both the smallest experimental unit and the number of seedlings used in general.
All require information is mentioned in the M&M and the suggested table having that information would be just repetition what were already described.
It is not clear where plants growing in one pot/tray/dish were used and where batches of plants were used. No information is provided about microbial contamination possible effects in root exudation experiment.
In the lines 343- it was mentioned that one plant per pot was grown and added a sentence describing where batches of plants were used (line 353-354). All trials were executed in well-equipped university laboratories under highest hygiene standards and diligence during the work. Therefore, to the best of our knowledge we could exclude any contamination.
Methodical references are only for barley (11,68).
We have added two more references (10 and 66) already cited in the manuscript, in which some other plant species were studied using the same exposure system. However, we have tested even more plant species using the same method and if all of them are cited than the number our references would increase considerably. Line 359
10 Elhakeem, A.; Markovic, D.; Broberg, A.; Anten, N. P. R.; Ninkovic, V. Aboveground mechanical stimuli affect belowground plant-plant communication. PLoS One 2018, 13 (5), e0195646. DOI: 10.1371/journal.pone.0195646.
66 Markovic, D.; Nikolic, N.; Glinwood, R.; Seisenbaeva, G.; Ninkovic, V. Plant responses to brief touching: A mechanism for early neighbour detection? PLoS One 2016, 11 (11), e0165742. DOI: 10.1371/journal.pone.0165742.
Eight out of 68 references belong to the authors of the article
See comment above.
Results
Age of the seedlings should be provided in the titles of the tables and the numbers of the plants used in separate replicates and the total number of plants should be provided
This is described in the methods sections. All seedlings were of uniform age when experiments started.
Discussions
Present study data are not compared with existing VOC data concerning selected species.
“In general, chlorophyll synthesis is not only dependent on numerous environmental factors like nu- trient, light, and water availability, but also consists of a series of enzymatic reactions that are sensitive to various biotic and abiotic factors” two parts of the sentence are talking about the same – I would reject such a phrase.
Thank you for this indication. We have rewritten this sentence in order to fulfil your requirement, lines 307-311.
“showed that variations in the chlorophyll content among coexisting species are significantly more influenced by the community composition than by site conditions like soil and weather, indicating a strong influence of plant–plant communication” might be very speculative interpretation due to limited experimental cases.
Thank you for this input. In our study, we found a strong influence of plant-plant-communication on results. The results of Li et al. (2018) underline our findings.
References
Should be more related to the species under investigation
In conclusion, present research is pure sterile laboratory experiment, which does not have relation to the reality either in agroecosystems or seminatural habitats. It may serve for government speakers to talk about Ambrosia threats.
To get a first insight how airborne VOCs of a plant species (irrespective if native or non-native weed or crop) influence neighboring plants it is essential to perform laboratory experiments under controlled conditions. Of course, it would be also interesting which role these interactions play in situ under natural growing conditions. When you are interested in more applied work on the influence of Ambrosia on the crops investigated, we would like to recommend you the following paper which was related to the present study:
Hall RM, Wagentristl H, Renner-Martin K, Urban B, Durec N, Kaul, H-P (2023). Extracts and Residues of Common Ragweed (Ambrosia artemisiifolia L.) Cause Alterations in Root and Shoot Growth of Crops. Plants 12, 1768. https://doi.org/10.3390/ plants12091768
https://www.mdpi.com/2223-7747/12/9/1768

Reviewer 3 Report
Manuscript title: Talking different languages: The role of plant–plant communication when an invader beats up a strange neighborhood.
This study has certain significance in plant–plant communication. However, revisions are necessary for the current version of the manuscript. Overall, I liked the basic idea of the manuscript. Intercropping and similar mixed cropping systems will become more and more important now that we are striving for input reduced cropping systems. However, there are several issues that I noted.
- It was hard to follow through the experimental set-up, despite having a figure (1) to illustrate it. It is unclear how the exposurewas counted.
The language is fine.
Author Response
Talking different languages: The role of plant–plant communication when an invader beats up a strange neighborhood.
This study has certain significance in plant–plant communication. However, revisions are necessary for the current version of the manuscript. Overall, I liked the basic idea of the manuscript. Intercropping and similar mixed cropping systems will become more and more important now that we are striving for input reduced cropping systems. However, there are several issues that I noted.
It was hard to follow through the experimental set-up, despite having a figure (1) to illustrate it.
This is our well known experimental set-up and already describer in previous papers (10, 11, 66 and 68 - references mentioned in this manuscript) and many other previous paper and widely accepted method to study volatile interaction between plants.
It is unclear how the exposure was counted.
Thank you for your comment. We are not quite sure, what you mean with “counted”.
- Seedlings of all crop species were exposed to airborne VOCs of Ambrosia (each pairing in 14 replications)
- Seedlings of Ambrosia were exposed to airborne VOCs of a kin or a crops species (each pairing in 14 replications)
- As control 14 plants of each species were grown without any neighbor (no exposure)
The exposure time was 21 days without any interference.
After that we did the measurements.

Round 2
Reviewer 2 Report
Main results should be briefly presented in the abstract. I am missing in the text information, provided for me: As control, 14 seedling of all plant species were grown without any neighbor (clean air - no airborne VOC information). It would be better to include such explanations in the text: „at the beginning of the exposure experiment all plant species were of the same age (first leaves fully developed). Subsequently, they were left for 21 days in the exposure chambers. The root experiment was similar. Seeds of all species were pre-germinated in petri dishes until the radicle and the cotyledones were visible. Then they were transferred to the tubes. Despite your laboratories are perfectly equipped, „21 days in the exposure chambers“ are enough for significant contamination, some assumptions should be added. It is generally accepted rule to include into tables and figures brief information without time consuming search/readings of all methods where details of time scales are scattered throughout all description. VOC are not detailed for examined speciesAuthor Response
We greatly value further comments and suggestions from the reviewer 2, which has enhanced our manuscript for readability and comprehension.
Main results should be briefly presented in the abstract.
We are of the opinion that the main results are well summarized in lines 21-26, in a clear and concise manner.
I am missing in the text information, provided for me: As control, 14 seedling of all plant species were grown without any neighbor (clean air - no airborne VOC information).
This information is missing in our manuscript and we are grateful to the reviewer for this suggestion. We have added text explaining exposure of control plants. See the lines 372-374.
It would be better to include such explanations in the text: „at the beginning of the exposure experiment all plant species were of the same age (first leaves fully developed). Subsequently, they were left for 21 days in the exposure chambers.
Thank you for this comment, now we included this information in the text, see the lines 374-376.
The root experiment was similar. Seeds of all species were pre-germinated in petri dishes until the radicle and the cotyledones were visible. Then they were transferred to the tubes.
We corrected this sentence as suggested. See in the lines 412-413.
Despite your laboratories are perfectly equipped, „21 days in the exposure chambers“ are enough for significant contamination, some assumptions should be added.
We don’t see how the plants can be significantly contaminated in our dual chamber exposure system with two separate cages. The system is specifically engineered to prevent all forms of contamination and guarantee only volatile interaction between the emitter plant situated in the induction cage and the receiver plant situated in the receiving cage (as shown in Figure 8). Clean air enters the induction cage, passes through the hole in separating wall into the receiving cage and is then vented outside the room (climate chamber) by a vacuum pump. The airflow in the system was tested using smoke. The air in the initial chamber fully changed after 2 minutes, as it passed from the emitting to the receiving chamber. The air was then vented outside the growth chamber. The air vented outside the room did not re-enter the exposure room, making it impossible for volatile organic compounds (VOCs) to contaminate any treatment in the exposure room. To prevent potential interaction between plants by root exudates, pots with emitting and receiving plants were placed in the separate petri dishes (this sentence added in the text). Please see the system description in line 362-369 and Figure 8.
In addition, it is unclear which type of contamination the reviewer wishes to address. All plants were subject to the same conditions, so any contamination would affect the entire experiment equally. It would be helpful to receive any assumptions from the reviewer regarding potential contamination, as we have conducted multiple experiments using the same set-up without any comments about contamination during the exposure period.
It is generally accepted rule to include into tables and figures brief information without time consuming search/readings of all methods where details of time scales are scattered throughout all description.
This table would be very small as there was only the post-germination phase until the first pair of leaves was visible and then the plants were exposed for 21 days. A table with 2 rows and 2 columns does not make any sense. Brief information about treatments etc. is necessary i. e. in field experiments with sowing, irrigating, fertilizing, harvesting etc., but not in this experiment.
VOC are not detailed for examined species
The objective of this study was to refrain from identifying any volatile compounds that might be accountable for the adaptation response of plants under exposure. Rather, our aim was to establish that volatiles emitted from undamaged plants have a significant impact on plant interactions and trigger adaptation responses in neighboring plants, but only when released in specific combinations. We aimed to demonstrate the role of ragweed volatiles in its invasiveness and crop adaptation strategy when these volatiles are detected, as well as the importance of VOCs and root exudates in kin recognition. Analysis of the VOCs profile of each species is part of an ongoing study in which we aim to identify the individual compounds or their mixture responsible for the observed effects.
As stated in the initial letter of justification, several VOCs from ragweed have already undergone analysis and been published.
Hall RM, Wagentristl H, Renner-Martin K, Urban B, Durec N, Kaul, H-P (2023). Extracts and Residues of Common Ragweed (Ambrosia artemisiifolia L.) Cause Alterations in Root and Shoot Growth of Crops. Plants 12, 1768. https://doi.org/10.3390/ plants12091768
https://www.mdpi.com/2223-7747/12/9/1768
Additional changes:
Line 17: we added “with crops”
Figure 7a: Root exsudate changed to Root exudates. Please see the new figure page 7.